# Dream to Adapt: Meta Reinforcement Learning by Latent Context Imagination and MDP Imagination

## Abstract

Meta reinforcement learning (Meta RL) has been amply explored to quickly learn an unseen task by transferring previously learned knowledge from similar tasks. However, most state-of-the-art algorithms require the meta-training tasks to have a dense coverage on the task distribution and a great amount of data for each of them. In this paper, we propose MetaDreamer, a context-based Meta RL algorithm that requires less real training tasks and data by doing meta-imagination and MDP-imagination. We perform meta-imagination by interpolating on the learned latent context space with disentangled properties, as well as MDP-imagination through the generative world model where physical knowledge is added to plain VAE networks. Our experiments with various benchmarks show that MetaDreamer outperforms existing approaches in data efficiency and interpolated generalization.

## 1 Introduction

Meta reinforcement learning has been widely explored to enable quick adaptation to new tasks by learning "how to learn" across a set of meta-training tasks. State-of-the-art meta-learning algorithms have achieved great success in adaptation efficiency and generalization. However, these algorithms rely heavily on abundant data of a large number of diverse meta-training tasks and dense coverage on the task distribution, which may not always be available in real-world. With insufficient meta-training tasks and training data, meta-learner can easily overfit and thus lack of generalization ability.

One major type of Meta RL is context-based method Rakelly et al. (2019), that learn latent representations of a distribution of tasks and optimizes the policy conditioned on the latent representations. This type of method allows off-policy learning and usually have rapid adaptation within several test-time steps. For these methods, the challenge of generalization is twofold. First, the encoder is required to infer unseen tasks properly, and second, the policy should interpret the unseen task representations and output optimal behaviors conditioned on latent task context.

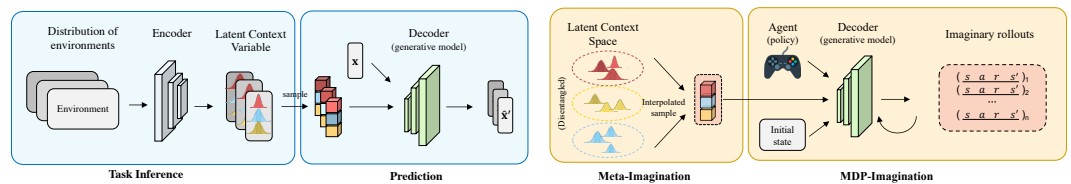

(a) Encoder-Generative model structure.   (b) Two types of imaginations in MetaDreamer

Figure 1: MetaDreamer schematic. We introduce meta-imagination and MDP-imagination in the context-based Meta RL framework to improve policy generalizability.

In this work we present MetaDreamer (as shown in Figure 1), a context-based MetaRL algorithm that improves policies' generalization ability without additional test-time adaptation through imagination. We base our method on three novel contributions. First, we design to learn an encoder that can represent tasks in a disentangled latent context space (Section 4.1), which enables efficient

and reasonable interpolation. Second, we integrate physics knowledge or specific domain information to the neural-network(NN)-based world model to better capture variants and common structure shared among a distribution of tasks (Section 4.2). Third, we introduce latent-context-imagination (meta-imagination) by sampling on the learned disentangled latent context space, and conditionally generate imaginary rollouts to augment training data (MDP-imagination) (Section 4.2). The training process for both the meta-imagination and MDP-imagination is described in Section 4.3.

## 2 RELATED WORK

**Meta Reinforcement Learning.** Current Meta RL algorithms can be categorized into three types: 1) context-based Meta RL Zhang et al. (2021)Mendonca et al. (2019)Rakelly et al. (2019)Kirsch et al. (2019)Gupta et al. (2018), as described in Section 1; 2) gradient-based Meta RL Finn et al. (2017)Nichol et al. (2018), which learns a policy initialization that can quickly adapt to a new task after few steps of policy gradient. These methods are usually data-consuming during both meta training and online adaptation ; 3) RNN-based Meta RL Wang et al. (2016)Duan et al. (2016b), preserving previous experience in a Recurrent Neural Network (RNN). This type of meta learner lacks of mathematical convergence proof and is vulnerable to meta-overfitting Finn et al. (2017). But recent model-based RNN Meta RL algorithms varibad Zintgraf et al. (2020), off-policy varibad Dorfman et al. (2020) show improved stability and generalization capability.

**Data augmentation and Imagination in Machine Learning.** Data augmentation has been widely used to improve image-based tasks with machine learning Wong et al. (2016)Zhu et al. (2017)Zhang et al. (2019) and meta-learning Ni et al. (2021)Yao et al. (2021)Khodadadeh et al. (2020), the techniques of which inspire imagination and are then applied in RL Hafner et al. (2019)Hafner et al. (2020), and Meta RL Kirsch et al. (2019)Lee & Chung (2021)Lin et al. (2020)Mendonca et al. (2020) to improve generalization. However, imagination in RL simply augments data for one task. In Meta RL, previous work generates imaginary tasks either with limited diversity Mendonca et al. (2020), or lack of control and interpretability Kirsch et al. (2019)Lee & Chung (2021)Lin et al. (2020). The most similar work to ours is LDM byLee & Chung (2021), generating imaginary tasks from latent dynamics mixture, but it generates imaginary tasks much less efficiently, lacking of control, focusing only on reward-variant MDPs, and involving more complex training.

**Physics-informed Machine Learning.** There has been a growing interest to integrate physics with machine learning. A systematic taxonomy of physics-informed learning includes: 1) knowledge types, e.g. natural scienceWang et al. (2017)Ren et al. (2018), world knowledgeXu et al. (2018)Ratner et al. (2016), domain expertiseKaplan et al. (2017)Ratner et al. (2016), etc.; 2) knowledge representation and transformation, e.g. differential equationsYang & Perdikaris (2018)Reinbold & Grigoriev (2019), rulesDiligenti et al. (2017)Stewart & Ermon (2017), constraintsStewart & Ermon (2017)Daw et al. (2017), etc.; 3) Knowledge integration, e.g. into the hypothesis spaceTowell & Shavlik (1994)Constantinou et al. (2016), to the training algorithmReinbold & Grigoriev (2019)Yang & Perdikaris (2018)Daw et al. (2017), etc. Especially, physics is shown to increase the capabilities of machine learning models to facilitate transfer learning in Goswami et al. (2020)Desai et al. (2021)Wang et al. (2021)Chen et al. (2021).

## 3 PRELIMINARY

**Formulation of Meta RL:** Following the traditional Meta RL formulations Finn et al. (2017), the goal of a meta-learner is to learn how to do quick adaptation through learning from meta-training tasks. We denote each task with $\mathcal{T}_i = (S, A, P_i, R_i, \gamma)$, where $S$ is the state space, $A$ is the action space, $P_i$ is the transition probability function, and $R_i$ is the reward function. A meta learner $\mathcal{M}$ with parameters $\theta$ can be optimized by:

$$\theta^* := \arg\max_\theta \sum_{i=1}^{N} \mathcal{J}(\phi_i, \mathcal{D}_i^{tr}), \quad \phi_i = f_\theta(\mathcal{D}_i^{ts}). \tag{1}$$

where $\mathcal{D}_i^{ts}$ is the test set of task $i$, $\mathcal{D}_i^{tr}$ is the training set of task $i$, $\mathcal{J}$ denotes the objective function (usually in the expected reward form). The meta learner, parameterized by $\theta$, takes the training set as input and learns a policy through the optimization process expressed in the above equation. Note

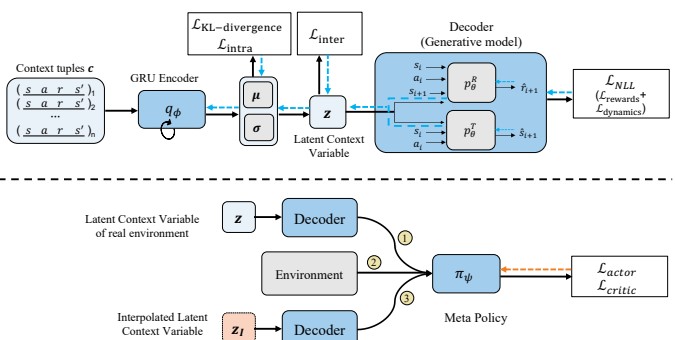

Figure 2: Components and training procedure of MetaDreamer. We disentangle the training of encoder-decoder(upper) and meta policy(lower). Black arrows outline the data flow, blue and orange arrows outline the gradient flow of encoder-decoder and meta policy respectively.

that the optimization process could include the learning of an adaptation operation $f_\theta$. Confronted with a new task $\mathcal{T}_i$, the meta-learner adapts from $\theta$ to $\phi_i$ by performing $f_\theta$ operation, which varies among different types of Meta RL methods.

**Algorithm of Meta RL:** We selected **PEARL** Rakelly et al. (2019), a prominent context-based, off-policy Meta RL algorithm, capturing features of current task in a latent probabilistic context variable $\mathbf{z}$ and doing adaptation by conditioning policy on the posterior context $\mathbf{z}$. PEARL has two key components: a probabilistic encoder $q(\mathbf{z}|\mathbf{c})$ for task inference, where $\mathbf{c}$ refers to context (at timestep $t$, context $\mathbf{c}_t = (\mathbf{s}_t, \mathbf{a}_t, r_t, \mathbf{s}_{t+1})$), and a task-conditioned policy $\pi(\mathbf{a}|\mathbf{s}, \mathbf{z})$. Due to PEARL's data efficiency and quick adaptation, we build our MetaDreamer on top of it.

## 4 METADREAMER

In this work, we build MetaDreamer as an off-policy context-based MetaRL algorithm, preserving good data efficiency and quick adaptation. The framework consists of an encoder for task inference, a decoder for data generation, and a meta policy to be trained with different task variants. The schematic of MetaDreamer is shown in Figure 1, and detailed training procedure in Figure 2.

Similar to PEARL, we extract information from the transition history and use the encoder to infer the value of latent context, which can be interpreted as latent task embeddings. These latent task embeddings are provided to task-conditioned policy learned via SAC to enable adapted behaviors.

One major difference between PEARL and our framework is that we decouple the task inference from the training of meta policy by reconstructing the tasks' MDPs in an unsupervised setup (similar to Bing et al. (2021)) with a newly-introduced decoder, which can work as a generative model. Another difference is that our training of the meta policy involves *meta-imagination* by sampling in the latent context space to generate imaginary tasks from the corresponding sampled latent vectors, and *MDP-imagination* by using the conditional decoder to generate imaginary rollouts.

### 4.1 TASK INFERENCE

**Encoder structure**. To enable adaptation, the encoder is supposed to extract task-related information from high dimensional input data, context $\mathbf{c}$, and represents the task with a latent context vector $\mathbf{z}$. Normally, researchers Rakelly et al. (2019)Wen et al. (2021) uses the permutation invariant property of a fully observed MDP and model the representation as a product of independent Gaussian factors. However, this inference architecture treats every context tuple with the same importance, which is not efficient in information gathering and can be problematic when informative tuples are very sparse. To overcome this problem, we employ a GRU to learn which data is important to keep or forget, and pass relevant information down the chain of sequential data. We recurrently feed in the context tuples and store our per-time-step posterior inference of the task in the GRU's hidden state.

**Disentangled Latent Context Space.** From the encoder, we can get low-dimensional latent representations (latent contexts) for different tasks. These latent representations constitute the latent context space, on which our following meta-imagination will be sampling from. However, previous context-based Meta RL methods learn a latent context space that can be convoluted and lacks phys-

ical interpretability. Major drawbacks of doing interpolation in such an unorganized latent context space include no control over the distribution of generated tasks, either too concentrated on a small range of the task domain or covering interpolated latent context space that doesn't have a reasonable mapping to the task space. Besides, the lack of physical interpretability prohibits customized study in task variants that interest us most.

In our structure, we enforce disentanglement on the latent context to perform efficient and maneuverable generation of imaginary tasks by following the beta-VAE techniques Higgins et al. (2016). As stated in Chen et al.Tokui & Sato (2021), a disentangled representation is not strictly defined, but can be described as one where a single dimension of the latent context are sensitive to changes in a single generative factor while being relatively invariant to changes in other factorsBurgess et al. (2018). A lot of work Higgins et al. (2016)Chen et al. (2018)Chen et al. (2016)Tran et al. (2017) have explored learning disentangled factors and we choose to leverage the $\beta$-VAE method Higgins et al. (2016) for disentanglement performance, training stability, and implementation complexity. The key idea of $\beta$-VAE is augmenting the original Variational Autoencoder (VAE) objective with a single hyperparameter $\beta$ (usually $\beta > 1$) that modulates the learning constraints on the capacity of the latent information bottleneck. It's suggested in Montero et al. (2020) that stronger constraints on the capacity of $\mathbf{z}$ is necessary to encourage disentanglement.

By employing $\beta$-VAE, we can learn a latent context space with following properties: 1) The latent context are disentangled; 2) The latent context only uses one active dimension for one generative factor, while leaving other dimensions invariant to any generative factor and close to normal distributions; 3) The latent context extracts minimal sufficient information, capturing all and only task-variants-relevant information. To be concise, all *generative factors* in this paper refer to the generative factors that are task-variant, excluding those invariant among all tasks.

## 4.2 IMAGINATION USING GENERATIVE MODELS

In MetaDreamer, we consider two types of imagination, *meta-imagination* and *MDP-imagination*, both for augmenting learning data but with some differences.

**Meta-imagination by sampling the latent context space.** Meta-imagination refers to the process of generating new tasks to augment the meta-training tasks, through sampling a set of imaginary (denoted with $\mathcal{I}$) latent context $\mathbf{z}^{\mathcal{I}} = \{\mathbf{z}^{\mathcal{I}(0)}, ..., \mathbf{z}^{\mathcal{I}(n)}\}$. Taking advantage of our learned disentangled context space, we can connect different features of tasks with corresponding latent context values inferred by the encoder. Suppose we have a set of meta-training tasks $\mathcal{T} = \{\mathcal{T}^{(0)}, ..., \mathcal{T}^{(N-1)}\}$ with dataset $\mathcal{D}_{\mathcal{T}}$ collected by interacting with corresponding environments. Performing task inference with the encoder, we can get a latent variable set $\mathbf{Z}^{\mathcal{T}} = \{\mathbf{z}^{\mathcal{T}^{(0)}}, ..., \mathbf{z}^{\mathcal{T}^{(N-1)}}\} = (Z_0 \times Z_1..... \times Z_{M-1})$, where $M$ is the dimension number of the latent context variable vector. Since not necessarily all components/dimensions of the latent context vector can be disentangled or has an interpretable generative factor representation, we define a linear injective mapping $f$ between the generative factor index and the latent context vector index, $f : \{0, 1, ..., K-1\} \to \{0, 1, 2, ..., M-1\}$, where $K$ is the number of generative factors and $K \leq M$.

Given the latent representation set $\mathbf{Z}^{\mathcal{T}}$, MetaDreamer applies the task interpolation separately on the disentangled latent variable space to get latent representations of some imaginary tasks:

$$Z_k^{\mathcal{I}} = \left\{ \left( \lambda z_{f(k),i-1} + (1-\lambda) z_{f(k),i} \right) : \quad i \in \{1, ..., I_k\}; \lambda = \frac{j}{D_k} + \epsilon, \quad j \in \{0, ..., D_k\} \right\} \quad (2)$$

where $\epsilon$ is a small noise, $D_k$ is representing interpolation density, and $I_k$ is the number of possible values for $k^{\text{th}}$ generative factor. Here $Z_k^{\mathcal{I}}$ contains all interpolated elements of $k^{th}$ dimension of $Z^{\mathcal{I}}$.

With the above latent context interpolation, we can generate interpolated tasks by combining samples on each generative factor, with notation $\mathbf{Z}^{\mathcal{I}} = (Z_0^{\mathcal{I}} \times Z_1^{\mathcal{I}}... \times Z_{K-1}^{\mathcal{I}})$. From the prospective of generative factors, we can divide interpolated tasks into three types: 1) all generative factors are sampled from the latent space of real tasks $\mathbf{Z}^{\mathcal{T}}$, which is a special case where $\lambda_k = 0$ or $1, \forall k$, but with different combinations from those in $\mathbf{Z}^{\mathcal{T}}$ so as to generate new tasks; 2) generative factors are hybrid combinations of interpolated and existing value of factors, where $\exists k$ s.t. $\lambda_k = 0$, or $1$, and $\exists j$ s.t. $\lambda_j \in (0, 1)$; 3)generative factors are all with interpolated values, where $\lambda_k \in (0, 1), \forall k$. This task interpolation process by manipulating on the disentangled latent context space gives us a very high level of interpretation and freedom to look into generalization domains that interest us most.

**MDP-imagination with physics-informed generative models.** MDP-imagination refers to the process of generating trajectories with the conditioned generative model to augment training data, similar to the imagination in previous studies like *Dreamer* Hafner et al. (2019) and *Dreamer-v2* Okada & Taniguchi (2022). The generative model $p_\theta$ consists of two parts: the reward model $p_\theta^R$ and the transition model $p_\theta^T$. Given a random initial state $s_0$ and policy $\pi$, the generative model can simulate imagined rollouts by recursively applying $(\hat{s}_{t+1}, \hat{r}_{t+1}) = p_{\theta, a \sim \pi}(\hat{s}_t, a_t, \mathbf{z})$. Then we get the imagined dataset $\tilde{\mathcal{D}} = \{\tilde{\mathcal{D}}_\mathcal{T}, \tilde{\mathcal{D}}_{\mathcal{T}^\mathcal{I}}\}$. It's worth noting that $\tilde{\mathcal{D}}_{\mathcal{T}^{(n)}}$ is the augmented dataset of a real task with latent context $\mathbf{z}^{(n)}$ by the generative model, in comparison the dataset $\mathcal{D}_{\mathcal{T}^{(n)}}$ is collected by interacting with real task environment $\mathcal{T}^{(n)}$.

Even previous work on meta-learning Khodadadeh et al. (2020) or reinforcement learning with purely image-based tasks Hafner et al. (2019), that involves imagination, have shown great success in computer vision domains, there are many circumstances where purely data-driven approaches find their limits. Doing imagination in Meta Reinforcement learning is highly demanding, which requires the generative model to condition on variant task embeddings and generate alike rollouts that reflect or preserve nature of the corresponding tasks. In addition to accurate and structured generation, generalization on unseen task context is also desired for the generative model considering the meta-imagination. Purely data-driven learning can be arduous, and thus there is a need to introduce extra knowledge for better learning.

Compared with image-based tasks, Meta RL and RL (not purely image-based) enjoy a good nature of interpretability from MDP's formulation. In MDP, the transition probability potentially involves some physics, intuition, or domain knowledge that may be known to us. Therefore, we deploy physics-informed machine learning, which incorporates additional knowledge into the machine learning pipeline, to better capture the real-world model and improve generalization. In this section, we alternatively use "physics-informed", "additional information", which should be more generally interpreted as knowledge from physics and other scientific disciplines. Physics-informed machine learning is a highly developed field and we recommend two comprehensive surveys Von Rueden et al. (2019) Willard et al. (2020) for readers to know about state-of-the-art development and categorization.

In our MetaDreamer, we introduce additional knowledge to transition approximation part in the generative model. Knowledge we consider introducing to the NN-based model falls into categories listed in Von Rueden et al. (2019). We then use the highway-merging task as an example to summarize possible ways to integrate knowledge with neural networks as follows:

- Distinguish different parts of the input vector (Figure 3b.ⓐ);
- Assign physical meanings to nodes in neural network (Figure 3b.ⓑ);
- Design the computation or value ranges of nodes in neural network (Figure 3b.ⓒ);
- Apply physics knowledge and replace NN-based model (Figure 3b.ⓓ);

The implementation of physics information with neural networks in Figure 3b is explained: a) The *state s* can be partitioned according to different vehicles every dimension is describing. b) We assign physical meanings to the output nodes, representing the acceleration of each vehicle. c) With the knowledge that a vehicle's acceleration usually falls in range of (-3, 3) m/s, we apply $3 \times$ `tanh()` as the output activation function for nodes mentioned in b). d) We apply the common physics law as $f_{\text{phy}}$: $p_{t+1} = p_t + v_t \Delta t$, and $v_{t+1} = v_t + a_t \Delta t$ where $p$ is x or y position, $v$ is velocity on x or y.

## 4.3 TRAINING

**Encoder-Generative model training.** Our encoder-generative model training follows $\beta$-VAE's formulation and approach. We setup a GRU encoder $q_\theta(\mathbf{z}|\mathbf{x})$ parameterized with $\theta$, and a physics-informed decoder $p_\phi(\mathbf{x}|\mathbf{z})$ consisting of a reward model $p_\theta^R$ and a transition function model $p_\theta^T$. We model the reward model as a pure neural network, while the transition model contains both physics knowledge and neural networks. The latent context is pre-defined with a redundant number of dimensions comparing with the number of generative factors.

The training objective is to maximize $\log p_\phi(\mathbf{x})$, the evidence lower bound objective (ELBO) of which is usually optimized on in practical training. Augmenting ELBO with a hyperparameter $\beta$,

we arrive at the objective function of $\beta$-VAE:

$$\mathcal{L}(\theta, \phi; \mathbf{x}, \mathbf{z}, \beta) = -\mathbb{E}_{q_\phi(\mathbf{z}|\mathbf{x})}[\log p_\theta(\mathbf{x}|\mathbf{z})] + \beta D_{KL}(q_\phi(\mathbf{z}|\mathbf{x})||p(\mathbf{z})) \qquad (3)$$

where $\phi, \theta$ parameterize the distributions of the encoder and decoder respectively. The input data $\mathbf{x}$ are samples from a task distribution parameterized by generative factors $\mathbf{z}$. The prior $p(\mathbf{z})$ is typically set to the normal distribution $\mathcal{N}(0, 1)$ and posterior $q_\phi(\mathbf{z}|\mathbf{x})$ distributions are parameterized as Gaussians with a diagonal covariance matrix.

Since we have a Gaussian prior, the first term in Equation 3, named reconstruction loss or negative log-likelihood loss ($\mathcal{L}_{NLL}$), becomes the mean squared errors (MSE) between ground truth and prediction of states and rewards:

$$\begin{aligned}
\mathcal{L}_{NLL}(\theta, \phi) &= -\mathbb{E}_{q_\phi(\mathbf{z}|\mathbf{x})}[\log p_\theta(\mathbf{x}|\mathbf{z})] \\
&= \alpha_T \cdot ||p_\phi^T(\hat{\mathbf{s}}_{t+1}|\mathbf{s}_t, \mathbf{a}, \mathbf{z}) - \mathbf{s}_{t+1}||^2 + \alpha_R \cdot ||p_\phi^R(\hat{r}_{t+1}|\mathbf{s}_t, \mathbf{a}, \mathbf{z}) - r_{t+1}||^2
\end{aligned} \qquad (4)$$

where $\alpha_T, \alpha_R$ are tunable hyper-parameters, weights, on state and reward reconstruction.

In MetaDreamer, task inference requires higher accuracy and the posterior collapse problem is even worse. We introduce two clustering losses related to (1) intra-cluster similarities and (2) iner-cluster similarities respectively to improve encoder-generative model's performance. Both cluster similarities are calculated with Euclidean distance, defined as $d_2(\mathbf{x}, \mathbf{y}) = \sqrt{\sum_{i=0}^{D}(x_i - y_i)^2}$, where $\mathbf{x}, \mathbf{y} \in \mathbb{R}^D$. Considering (1), we encourage smaller intra-clusters distance, meaning that the inferred latent context with different trajectories of the same task are more similar. This encourages the encoder to extract task-variant-relevant factors while leaving out other information contained in trajectories, thus improving the accuracy and same-task-wise consistency of the inference. As for (2), we penalize too small inter-cluster distance. Intuitively, encouraging bigger inter-cluster distance helps with task-variant discovery and transmitting information from input to the latent context posterior. In this case, however, the KL divergence term in Equation 3 will also be encouraged to increase, possibly leading to poor disentanglement. Therefore, we simply penalize it when the inter-cluster distance of two tasks is below a threshold that these two tasks can be viewed as indistinguishable. The cluster loss is formulated as:

$$\mathcal{L}_{cluster}(\phi) = \alpha_{c1}\mathcal{L}_{intra}(\phi) + \alpha_{c2}\mathcal{L}_{inter}(\phi)$$

$$\mathcal{L}_{intra}(\phi) = \sum_{i=0}^{N-1} \frac{1}{K^{(i)}} \sum_{k=0}^{K^{(i)}} d_2(\mathbf{z}^{(i),k}, \bar{\mathbf{z}}^{(i)}); \mathcal{L}_{inter}(\phi) = \sum_{i_1=0}^{N-1} \sum_{i_2=i_1+1}^{N-1} \text{clip}(\sigma - d_2(\bar{\mathbf{z}}^{(i_1)}, \bar{\mathbf{z}}^{(i_2)}), 0, \sigma)$$

$$\text{where} \quad \bar{\mathbf{z}}^{(i)} = \frac{1}{M^{(i)}} \sum_{m=0}^{M^{(i)}} \mathbf{z}^{(i),m}, \quad \forall i = 0, ..., N-1$$

$$(5)$$

where $\sigma$ is the pre-defined minimal inter-cluster distance threshold, $\alpha_{c1}, \alpha_{c2}$ are tunable hyperparameters, $N$ represents the number of tasks, and for each task $(i)$, the losses are computed with $M^{(i)}$ trajectories.

Summarizing all losses, the encoder-generative model is trained with the overall objective function:

$$\mathcal{L}(\theta, \phi; \beta, \alpha_T, \alpha_R, \alpha_{c1}, \alpha_{c2}) = \mathcal{L}_{NLL}(\theta, \phi; \alpha_T, \alpha_R) + \beta \cdot \mathcal{L}_{KL}(\theta) + \mathcal{L}_{cluster}(\phi; \alpha_{c1}, \alpha_{c2}) \qquad (6)$$

**Meta policy training.** The policy training is using the soft actor-critic algorithm, an off-policy method based on the maximum entropy RL objective. Thus our policy training preserves good data efficiency and stability. During policy training, the gradients are not back-propagated through the encoder-decoder network. The critic and actor loss are written as:

$$\mathcal{L}_{critic} = \mathbb{E}_{\substack{(\mathbf{s},\mathbf{a},r,\mathbf{s}')\sim\mathcal{B} \\ \mathbf{z}\sim q_\phi(\mathbf{z}|\mathbf{c})}}[Q_\theta(\mathbf{s}, \mathbf{a}, \bar{\mathbf{z}}) - (r + \bar{V}(\mathbf{s}', \bar{\mathbf{z}}))]^2. \qquad (7)$$

where $\bar{V}$ is target value network and $\bar{\mathbf{z}}$ means that gradients are not back-propagated through network.

$$\mathcal{L}_{actor} = \mathbb{E}_{\substack{\mathbf{s}\sim\mathcal{B}, \mathbf{a}\sim\pi_\theta \\ \mathbf{z}\sim q_\phi(\mathbf{z}|\mathbf{c})}}\left[ D_{KL}\left( \pi_\theta(\mathbf{a}|\mathbf{s}, \bar{\mathbf{z}}) \,||\, \frac{\exp\left(Q_\theta(\mathbf{s}, \mathbf{a}, \bar{\mathbf{z}})\right)}{\mathcal{Z}_\theta(\mathbf{s})} \right) \right]. \qquad (8)$$

where the partition function $\mathcal{Z}_\theta$ normalizes the distribution.

The data used to infer $q_\phi(\mathbf{z}|\mathbf{c})$ is $\mathbf{c}$, sampling from the most recently collected data batch, while the actor and critic are trained with data from the entire data buffer $\mathcal{B}$. As shown in Figure 2, there are three types of data according to how they're generated: 1) performing policy $\pi_\psi$ conditioned on latent context $\mathbf{z}$ of a real task with its corresponding real environment (hereinafter referred to as R); 2) policy $\pi_\psi$ conditioned on $\mathbf{z}$ of a real task with its conditioned generative model $p_\theta$ (IR); 3) policy $\pi_\psi$ conditioned on $\mathbf{z_I}$ of an imagined task with its conditioned generative model $p_\theta$ (I).

## 5 EXPERIMENTS AND RESULTS

In this section, we conduct experiments to test and evaluate the performance of MetaDreamer. We first investigate the properties of MetaDreamer on two versions of *highway-merging* experiment, including the encoder's effectiveness, the decoder's generation quality, and the meta-policy's performance. We then validate the generalization of our algorithm by employing on four MuJoCo continuous control tasks widely used in the meta-RL literature. We show that MetaDreamer has better generalization capability to unseen tasks, than many state-of-the-art meta-learning methods, by doing meta-task augmentation for meta-policy training. In the appendix, we include more experimental designs, evaluation metrics and ablation studies.

### 5.1 AUTONOMOUS DRIVING EXPERIMENT.

We modify a classical self-driving scenario, highway merging Leurent (2018), into a simulator suitable for Meta RL testing by considering variants that can reflect complex real driving environments. As shown in Figure 8, the ego vehicle (in green) is expected to take a mandatory lane change to

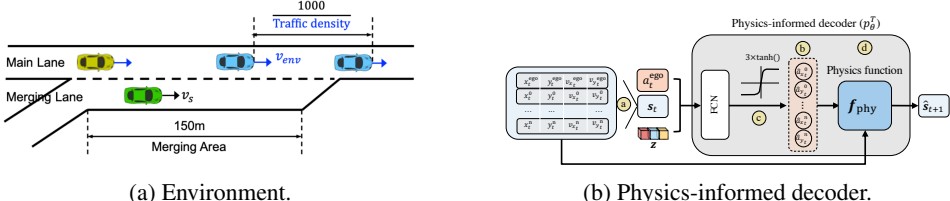

(a) Environment.                    (b) Physics-informed decoder.

Figure 3: *Highway-merging*. The ego vehicle (in green) is expected to learn a merging policy that can adapt quickly to different driving environments, considering task-variant generative factors: traffic speed, and longitudinally interactive model's parameters.

merge into the main lane before the merging area ends. Surrounding vehicles on the main lane follow the Intelligent Driver Model (IDM) Treiber et al. (2000) as longitudinal dynamics and ignore lateral dynamics in this merging area. Before the ego vehicle successfully merges, the vehicle right behind it (in yellow) follows a longitudinally interactive model. We use similar MDP configurations in Leurent (2018). Specific MDP configurations are described in the appendix. We design two complexity levels of highway-merging task with different longitudinally interactive models:

***Highway-Merging-v0*** (simple version): the yellow vehicle uses a proportional interactive model that follows: $a_p = p \cdot a_{ego}$, where $a_{ego}$ is ego vehicle's acceleration. To mimic different interactive styles, we consider $p$ as a variant with $p \in [-1, 1]$. Generally speaking, vehicles with higher positive $p$ tend to more opponents while those with smaller negative $p$ behave more cooperatively.

***Highway-Merging-v1*** (hard version): use the Hidas' interactive model Hidas (2005), among the parameters of which we consider two as task variants: 1) Maximum speed decrease $Dv \in [5, 15]$ (mph); 2) Acceptable deceleration rage $b_f \in [1.0, 3.5]$ (m/s$^2$)

We first illustrate the encoder's effectiveness on *highway-merging-v0* in Figure 5, plotting out $\mathbf{z}_i$-$\mathbf{g}_j$ graphs: how the value of the $i$ dimension of inferred latent context $\mathbf{z}_i$ changes to the change of generative factor $\mathbf{g}_j$ (e.g. traffic speed, and proportional values $p$). For each generative factor, we pick four different values in the plotting. It is shown that following $\beta$-VAE, MetaDreamer learns to infer disentangled latent context and only activate necessary dimensions.

Then we investigate the generation quality of the decoder, including episode generation (MPD imagination) and task interpolation (meta imagination). To give an intuitive illustration of the generation, we deploy our algorithm on *Highway-Merging-v0*, whose variant, $p$ value, can be visualized by the ratio of values on colorful lines (rear vehicle's acceleration profile) to corresponding values on the black line (ego vehicle's acceleration profile).

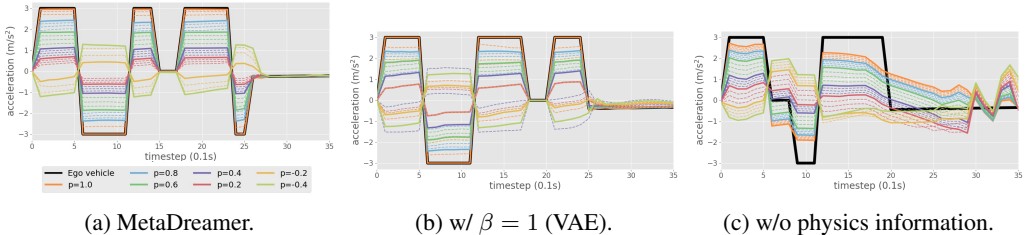

(a) MetaDreamer.            (b) w/ $\beta = 1$ (VAE).            (c) w/o physics information.

Figure 4: Illustration of interpolated tasks. Each plot shows acceleration profiles of the rear surrounding vehicle. Black lines are the ego vehicle's accelerations, colorful solid lines are predictions by the generative model on real tasks (theoretically should be $p$-proportional to black lines, representing MDP imagination), and dash lines are imagination with interpolated latent context between tasks with nearby $p$ value (meta imagination).

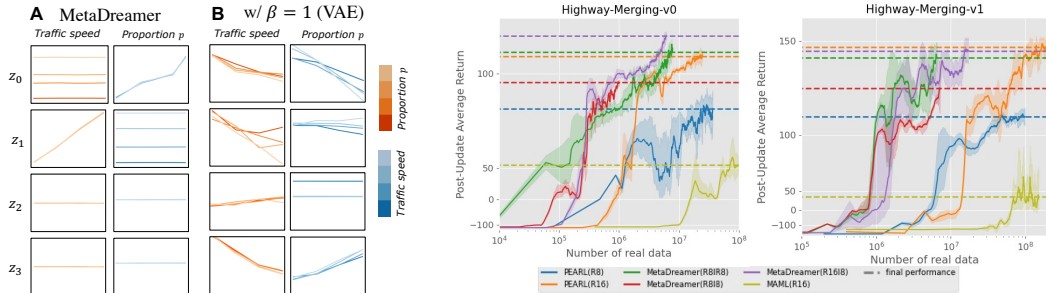

Figure 5: Latent context learnt by a MetaDreamer(with $\beta = 5$), and VAE (with $\beta = 1$). Each line represents a task with the same value on the generative factor in that color. We are plotting this figure in reference of Fig. 7 in Higgins et al. (2016)

Figure 6: Comparison of meta policy's training and adaptation to meta-testing performance. Post-adapted policy's episodic return on meta-testing tasks during the meta-training phase are illustrated. Each learning curve shows the mean (solid lines) and variance (shades) of three independent runs.

MPD imagination quality can be evaluated by the shape of the acceleration profile (colorful solid lines): the more consistent with the shape of ego vehicle's acceleration profile (black line), the better the MPD imagination quality is. In Figure 4a and 4b, we can observe more consistent ratios of colorful solid lines to the black line compared with Figure 4c, indicating better MPD imagination with the physics-informed generator. On the other hand, the meta-imagination quality can be investigated by the acceleration profile of generation by using the interpolated context vectors (colorful dash lines). In Figure 4a and 4c, dash lines are more distinguishable from each other and have denser and more regularized coverage in between solid lines, while in Figure 4b dash-lines are observed with a lot of overlapping and disordering. In general, Figure 4 intuitively presents that MetaDreamer has better MPD generation accuracy and interpolated meta-imagination.

With a well-trained encoder-generative model, we can evaluate the effectiveness of imagination in improving policy learning. We compare with two state-of-the-art Meta RL baselines: a context-based off-policy algorithm PEARL Rakelly et al. (2019), and a gradient-based model-agnostic meta-learning algorithm MAML Finn et al. (2017). Besides, we also compare policies trained with different sources and sizes of data, denoted by the letter and number following the algorithm's name (e.g. MetaDreamer(R8IR8) denotes using MetaDreamer algorithm to train meta-policy using rollouts of 8 real tasks and 8 imaginary real tasks).

From the meta-learning curves in Figure 6, we analyze the generalization capability of MetaDreamer by measuring its performance on meta-testing tasks as a function of the total number of samples used during meta-training. We use PEARL(R8) and PEARL(R16) as two important baselines to evaluate what types of imagination and how they contribute. By doing two pairs of comparison, MetaDreamer(R8I8) vs PEARL(R8), and MetaDreamer(R16I8) vs PEARL(R16), we can observe the outperformance of MetaDreamer, even though MetaDreamer(R16I8)'s superiority is less obvious. This is because the benefits of meta-imagination highly depend on the property of a task and the potential improvements based on available real data. Then if comparing the data efficiency again on these two pairs, we can observe 100x in R8 pair, and 10x in R16 pair, less data used to reach the same level of post-adaptation performance. Further on, we observe similar final performance comparing MetaDreamer(R8IR8) and PEARL(R16), indicating that augmenting data through MDP-imagination for environments that we have data access to is one approach to improve data efficiency.

## 5.2 OTHER EXPERIMENTS

In addition to the *Highway-Merging* task described above, to show the good extensibility of our algorithm, we present the experimental results of four commonly used meta-RL benchmark tasks used in existing meta-RL literature Rakelly et al. (2019)Finn et al. (2017)Yu et al. (2020). The details of each environment are described in the appendix.

In Figure 7, we can observe that MetaDreamer obviously outperforms PEARL on the *Walker-2D* experiment with regard to the post-update performance, and shows similar adaptive capability on the other three environments while using 10-100x less real data to (for a simple and fair comparison, we use the number of real data required when the post-update average return curve of MetaDreamer reaches the final performance line of PEARL as the comparison metric). These robotics experiments show that MetaDreamer significantly improves data efficiency in all environments compared to both on-policy and off-policy methods, and could improve the capability to do fast adaptation under certain environment distributions.

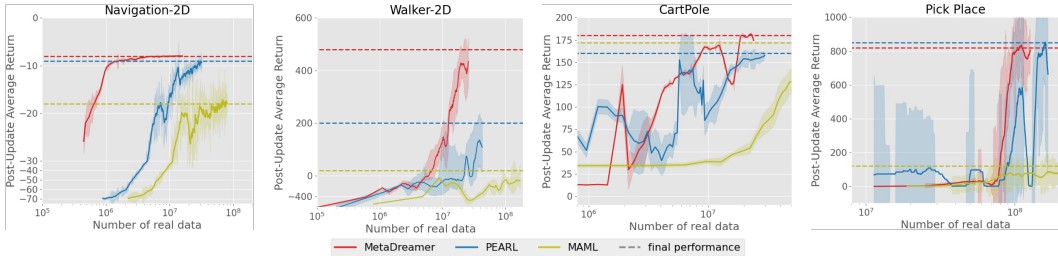

Figure 7: Comparison of meta policy's adaptation to meta-testing tasks of robotics environments.

## 6 CONCLUSION

In this paper, we have presented MetaDreamer, a novel off-policy context-based Meta RL algorithm that improves data efficiency and generalization capability by two types of imagination: *meta-imagination* can improve policy's generalization by augmenting training tasks with imaginary tasks through efficient sampling on the learned disentangled latent context space, and *MDP-imagination* can improve data efficiency by augmenting training data for real tasks. With a real generalization problem in autonomous driving and four common benchmark tasks, we evaluate and illustrate the encoder's task inference and a physics-informed generative model's generation performance, and validate the meta policy's better generalization and data efficiency.

In future work, we aim to further improve policy's generalization by generating extrapolated imaginary tasks, which exerts more challenges on the generative model's extrapolation capability. There are many interesting work that could provide theoretical and technical foundations Besserve et al. (2021)Takeishi & Kalousis (2021). On the other hand, considering non-parametric variants can greatly extend meta-learner's capabilities. This is still an open question Bing et al. (2021)Bing et al. (2022)Atkeson (1997)Goo & Niekum (2020) and how to take advantage of imagination in non-parametric meta-learning is barely explored.

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

## 7 APPENDIX

### 7.1 EXPERIMENT DETAILS

#### 7.1.1 *Highway-merging*

This *Highway-merging* experiment is modified from an OpenAI-Gym-based simulator Leurent (2018). For both easy version (*v1*) and hard version (*v2*), we use similar MDP configurations in Leurent (2018). The MDP configuration is described as follows:

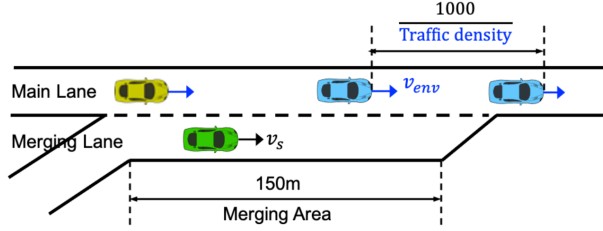

Figure 8: *Highway-merging* Environment.

The reward function $R(s, a)$ consists of four terms: velocity-related reward $R_v$ (encouraging keeping at a speed similar to the environmental average speed), merging-related reward $R_m$ (encouraging a safer merging gap to the front vehicle and to the rear vehicle, as well as higher merging efficiency),

| Dim. | Continuous Observation Space |
|------|------------------------------|
| 0-3 | $x$ lateral position, $y$ position to the end of merging, and $v_x, v_y$ velocity of ego vehicle |
| 4-7 | $\Delta x, \Delta y$ relative position and $\Delta v_x, \Delta v_y$ relative velocity of first front vehicle |
| 8-11 | $\Delta x, \Delta y$ relative position and $\Delta v_x, \Delta v_y$ relative velocity of second second front vehicle |
| 12-15 | $\Delta x, \Delta y$ relative position and $\Delta v_x, \Delta v_y$ relative velocity of first rear vehicle |
| 16-19 | $\Delta x, \Delta y$ relative position and $\Delta v_x, \Delta v_y$ relative velocity of second rear vehicle |

| Dim. | Range | Dim. | Range |
|------|-------|------|-------|
| $x$ | [0,12] | $\Delta x$ | [-10,10] |
| $y$ | [0,150] | $\Delta y$ | [-150,150] |
| $v_x$ | [0,50] | $\Delta v_x$ | [-20,20] |
| $v_y$ | [0,10] | $\Delta v_y$ | [-10,10] |

| Index | Discrete Action Space |
|-------|----------------------|
| 0 | idel |
| 1 | change to the left lane |
| 2 | change to the right lane |
| 3 | accelerate at 1.5m/s$^2$ |
| 4 | decelerate at -1.5m/s$^2$ |

Table 1: Observation space of *Highway-merge*. The observation will be normalized on each dimension before being fed into neural networks.

Table 2: Action space of *Highway-merge*

crash-related reward $R_c$ (punishing on a crash or a nearly crash) and action-related reward $R_a$ (punishing on acceleration/deceleration and lane-changing cost):

$$R_v = \text{clip}(\frac{|v - v_{env}|}{10}, 0, 1)$$

$$R_m = \frac{\text{relu}(s^*(v_{\text{ego}}, \Delta v_{\text{ego}}) - s_{\text{ego}})}{s^*(v_{\text{ego}}, \Delta v_{\text{ego}})} + \frac{\text{relu}(s^*(v_{\text{rear}}, \Delta v_{\text{rear}}) - s_{\text{rear}})}{s^*(v_{\text{rear}}, \Delta v_{\text{rear}})} + 20 \times (\dot{a}_{rear} < 0)$$

$$R_c = \begin{cases} -50 & \text{, if crashed} \\ 0 & \text{, otherwise} \end{cases}, \quad R_a = \begin{cases} -1, & \text{if } a = 1 \text{ or } 2 \\ -0.2, & \text{if } a = 3 \text{ or } 4 \\ 0, & \text{otherwise} \end{cases} \tag{9}$$

with $s^*(v_i, \Delta v_i), v_i, s_i$ is defined the same in the Intelligent Driving Model Treiber et al. (2000).

### 7.1.2 *Navigation-2D*

This experiment is used in Finn et al. (2017), where a point agent is desired to move to the goal positions in 2D space. The MDP configurations are defined as:

| Dimension | Continuous Observation Space | Range |
|-----------|------------------------------|-------|
| 0 | agent position $x$ | [-4,4] |
| 1 | agent position $y$ | [-4,4] |

Table 3: Observation space of *Navigation-2D*.

| Dimension | Continuous Action Space | Range |
|-----------|-------------------------|-------|
| 0 | agent velocity $\Delta x$ | [-1,1] |
| 1 | agent velocity $\Delta y$ | [-1,1] |

Table 4: Action space of *Navigation-2D*.

The reward is defined as: $R = -\sqrt{(x - x_{goal}^2 + (y - y_{goal})^2)}$ (the agent gets a higher reward if getting closer to the goal position). Episodes terminate at the horizon of $H = 100$. In this experiment, we consider the variants to be the goal position: $x_{goal} \in [-2, 2], y_{goal} \in [0, 2]$ to

consist of a set of tasks (illustrated in Fig.9, and the physics-informed decoder structure is shown in Figure 11a.

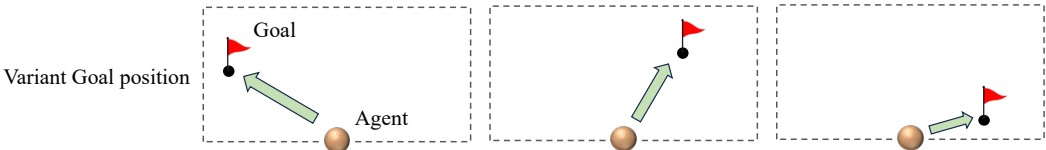

Figure 9: *Navigation-2D* experiment's variants.

### 7.1.3 *CartPole*

This is a classical OpenAI Gym-based control problem, which is with the goal to keep the cart pole balanced by applying appropriate forces to a pivot point. The dynamics of this system are fully described in Florian (2005). The MDP configurations are defined as:

| Dimension | Continuous Observation Space | Range |
|---|---|---|
| 0 | cart position $x$ | [-4.8,4.8] |
| 1 | cart velocity $\dot{x}$ | [-inf, inf] |
| 2 | pole flip angle $\theta$ | [-0.418, 0.418] |
| 3 | pole flip angle rate $\dot{\theta}$ | [-inf, inf] |

Table 5: Observation space of *CartPole*.

| Index | Discrete Action Space |
|---|---|
| 0 | push to the left |
| 1 | push to the right |

Table 6: Action space of *CartPole*

The reward function is designed to discourage deviation from the balanced position (small pole flip angle and angle rate, and slow movement of the cart):

$$R = 2 - 0.3 \times \tanh(||x||) - 0.2 \times \tanh(||\dot{x}||) - 0.3 \times \tanh(||\theta - \theta_{thrd}||) - 0.2 \times \tanh(||\dot{\theta}||) \quad (10)$$

Episodes terminate if the pole angle exceeds the threshold $\theta_{thrd}$ or at the horizon of $H = 100$. We consider the variants: gravity $g \in [0.1, 2] \times 9.8$ N/kg, force $|f| \in [5, 15]N$ to consist of a set of tasks (as illustrated in Fig. 10), and the physics-informed decoder structure is shown in Figure 11b.

### 7.1.4 *Walker-2D*

We also study on high-dimensional locomotion task with the MuJoCo 2D walker, which is a two-legged figure and learns to make coordinate both sets of feet, legs, and thighs to move by applying torques on the six hinges connecting the six body parts. The MDP configuration is the same to Duan et al. (2016a), but we design a more challenging meta-learning version by considering two variants: gravity $g \in [0.1, 2] \times 9.8$ N/kg, and desired moving velocity $v_{goal} \in [-3, 3]$ m/s (positive means forward and negative means backward).

The reward function consists of three parts: health-related reward $R_{health}$ (receives a fixed reward of value every timestep that the walker is alive), velocity-related reward $R_v$, and control-related reward $R_c$(encouraging less torque usage to perform the movement):

$$\begin{aligned} R_v &= (v - v_{goal})^2 \\ R_c &= F_t^2 + F_l^2 + F_f^2 \end{aligned} \quad (11)$$

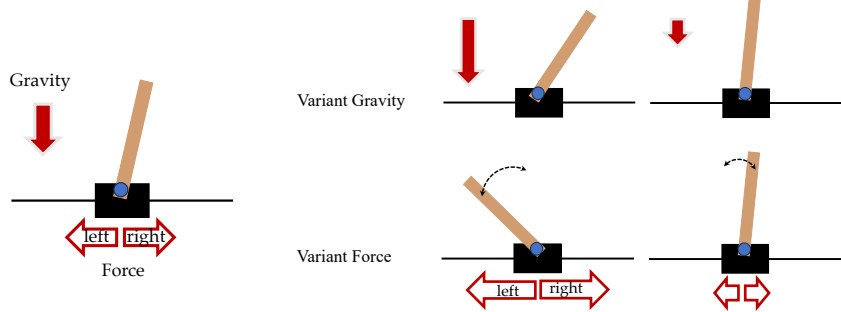

Figure 10: *CartPole* experiment's variants

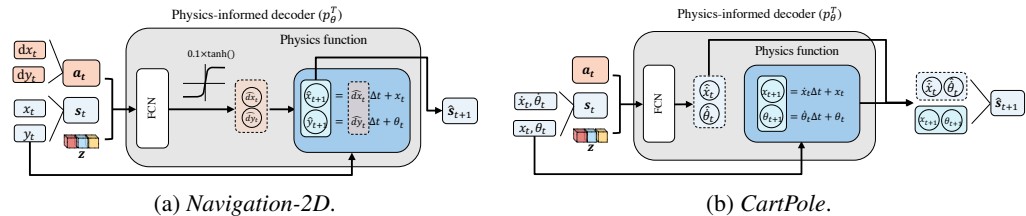

(a) *Navigation-2D.*                    (b) *CartPole.*

Figure 11: Physics-informed generative model's structures for **Navigation-2D** and **CartPole**.

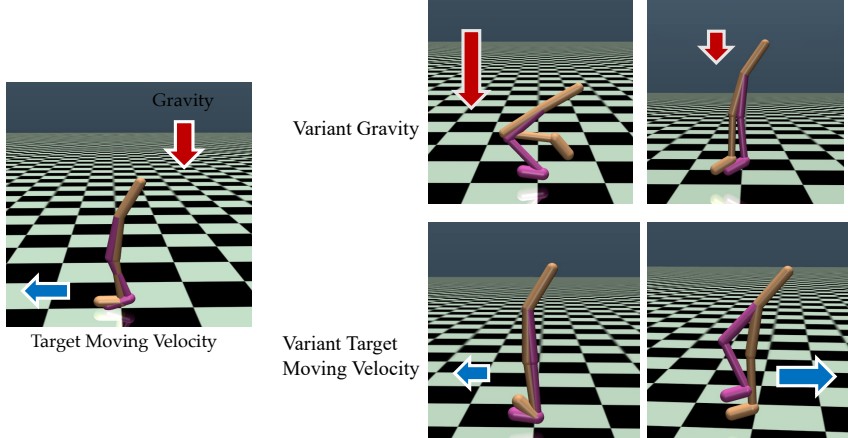

Figure 12: *Walker-2D* experiment's variants

### 7.1.5 *Pick-Place*

This experiment is chosen from the benchmark Meta-World Yu et al. (2020), which is with the goal to pick and place a puck to a goal position in the 3D space. We use the same MDP configuration in Yu et al. (2020) and constitute a set of different tasks by considering different goal positions ($x \in [-0.3, 0.3]$). Note that unlike **MT1** setting in Yu et al. (2020), the goal positions are not provided in the observation, as to test the ability of algorithms on generalization.

The reward function consists of two parts: grip-related reward $R_g$ (gets a high reward if the gripper holds the puck), and placement-related reward $R_p$ (gets a high reward if the puck is placed within a nearby area to the goal position)

The transition probability functions in **Walker-2D** and **Pick-Place** neither include objective motivation (like the acceleration in **Highway-merge**), nor have physics parts that can be easily seperated from the whole model. Therefore, we don't use the physics-informed decoder in these two experiments but focus on the effectiveness of two types of imagination.

| Dimension | Continuous Observation Space | Range |
|---|---|---|
| 0 | height of hopper $z$ | [-inf,inf] |
| 1 | angle of the top $y$ | [-inf,inf] |
| 2/5 | angle of the thigh joint (left/right) | [-inf,inf] |
| 3/6 | angle of the leg joint (left/right) | [-inf,inf] |
| 4/7 | angle of the foot joint (left/right) | [-inf,inf] |
| 8 | speed of the top $v_x$ | [-inf,inf] |
| 9 | speed of the top $v_z$ | [-inf,inf] |
| 10 | angular velocity of the top $\omega$ | [-inf,inf] |
| 11/14 | angular velocity of the thigh (left/right) $\omega_t$ | [-inf,inf] |
| 12/15 | angular velocity of the leg (left/right) $\omega_l$ | [-inf,inf] |
| 13/16 | angular velocity of the foot (left/right) $\omega_f$ | [-inf,inf] |

Table 7: Observation space of *Walker-2D*.

| Dimension | Continuous Action Space | Range |
|---|---|---|
| 0/3 | Torque applied on the thigh rotor (left/right) $F_t$ | [-1,1] |
| 1/4 | Torque applied on the leg rotor (left/right) $F_l$ | [-1,1] |
| 2/5 | Torque applied on the foot rotor (left/right) $F_f$ | [-1,1] |

Table 8: Action space of *Walker-2D*.

## 7.2 ENCODER-GENERATIVE MODEL EVALUATION

### 7.2.1 EVALUATION METRIC

The encoder-generative model is the key part in MetaDreamer, which grounds our understanding of tasks, and augments meta-learning tasks through imagination as well. To evaluate the performance of our encoder-generative model, we design five metrics to systematically evaluate the performance of learned encoder-generative model, which are defined as follows:

**Disentanglement score.** We compute the disentanglement metric score in a way similar to Higgins et al. (2016), with some modification and described in the appendix. The higher in percentage, the more disentangled the learned latent representations are.

**Intra-cluster variance.** This metric is the intra-cluster loss in Equation 5 and evaluates the same-task-wise consistency of task inference.

**Reconstruction error.** This metric is the $\mathcal{L}_{NLL}$ in Equation 4, indicating likeness of the generative model to the real world from the simulated time-step level.

**Speculated Factor values of Imaginary trajectories (SFI) error.** We infer the speculated values $\hat{\mathbf{g}}$ of generative factors for each time-step tuple $(s, a, s', r)$ in an imaginary trajectory $p(\tilde{\tau}|q(\mathbf{z}|\tau))$, where $\tau$ is a collected trajectory in real environment, and $\tilde{\tau}$ is a imaginary trajectory. SFI error is represented in form $(\text{mean}(\{|\hat{\mathbf{g}}_k - \mathbf{g}_k|\}_k) \pm \text{var}(\{|\hat{\mathbf{g}}_k - \mathbf{g}_k|\}_k))$, indicating the absolute bias and variance of MDP-imagination from the trajectory level.

**Speculated latent Context of Imaginary tasks (SCI) error.** The computation of SCI error is following: $\text{mean}(\{|\hat{\mathbf{z}} - \mathbf{z}|\}) \pm \text{var}(\{\hat{\mathbf{z}}\})$, by doing task inference on imaginary trajectories of an interpolated latent context, denoted as $q(\hat{\mathbf{z}}|p(\tilde{\tau}|\mathbf{z}))$, where $\mathbf{z}$ is a randomly sampled, interpolated latent context. The SCI error indicates the level of alignment of MDP generation and task inference considering meta-imagination.

### 7.2.2 EVALUATION RESULTS

We evaluate the learned encoder-generative model with our proposed evaluation metrics in Table 11. All experiments show satisfying disentanglement property (with a nearly 100% disentanglement score), task-consistent inference (small intra-cluster variance), accurate reconstruction (nearly

| Dimension | Continuous Observation Space | Range |
|---|---|---|
| 0 | robot arm position $x$ | [-0.2,0.2] |
| 1 | robot arm position $y$ | [0.5,0.8] |
| 2 | robot arm position $z$ | [0.05,0.3] |
| 3 | gripper paddle open angle $d$ | [0,1] |
| 4 | puck position $x_p$ | [-0.2,0.2] |
| 5 | puck position $y_p$ | [0.5,0.8] |
| 6 | puck position $z_p$ | [0,0.3] |
| 7-10 | puck quaternion | [-1,1] |

Table 9: Observation space of *Pick-Place*.

| Dimension | Continuous Action Space | Range |
|---|---|---|
| 0 | robot arm movement $\Delta x$ | [-1,1] |
| 1 | robot arm movement $\Delta y$ | [-1,1] |
| 2 | robot arm movement $\Delta z$ | [-1,1] |
| 3 | gripper paddle distance apart$\Delta d$ | [-1,1] |

Table 10: Action space of *Pick-Place*.

0 reconstruction error for experiments with physics information, and small enough for those without physics information), authentic imagination and good alignment of MDP-imagination and task inference (small SFI and SCI error)

| Experiment | Disentanglement score | Intra-cluster variance | Reconstruction error | SFI error | SCI error |
|---|---|---|---|---|---|
| *Navigation-2D* | $98.4 \pm 0.5\%$ | $0.27 \pm 0.15$ | $0.11 \pm 0.06$ | $0.13 \pm 0.05$ | $0.36 \pm 0.14$ |
| *CartPole* | $96.3 \pm 0.7\%$ | $0.31 \pm 0.23$ | $0.13 \pm 0.14$ | $-$ [1] | $0.53 \pm 0.21$ |
| *Walker-2D* | $90.7 \pm 4.7\%$ | $0.41 \pm 0.17$ | $1.22 \pm 0.17$ | $0.25 \pm 0.14$ [2] | $0.44 \pm 0.18$ |
| *Pick-Place* | $93.4 \pm 5.3\%$ | $0.38 \pm 0.22$ | $1.09 \pm 0.08$ | $0.31 \pm 0.20$ | $0.72 \pm 0.25$ |

Table 11: Evaluation of encoder-generative model.

[1] The variant generative factors are too complex to speculate from state transitions, so we don't evaluate the SFI error for this experiment.

[2] The generative factor, gravity, is too complex to speculate, so we only computate the SFI error on the generative factor, desired moving velocity.

### 7.2.3 INTERPOLATION VISUALIZATION

The interpolation performance is hard to be directly evaluated, but we manage to visualize the interpolation results of the following two experiments.

***Navigation-2D.*** The considered variant, goal positions, can be visualized by plotting out the reward heatmap (Figure 14). Because the reward function is proportional to the $l2$ distance to the goal position, each figure's brightest area represents the goal position of that corresponding task. It can be observed that $z_0$ and $z_1$ represent the goal position changing on two diagonal lines respectively, while $z_2$ and $z_3$ are inactive in this experiment.

***Walker-2D.*** Though the reward consists of several components, velocity reward is predominant. Therefore, the considered variant, desired moving velocities, can be approximately visualized by plotting out the reward heatmap (Figure 15). The brightest area corresponds to the approximate desired moving velocity. We can observe that a more positive $z_0$ controls a larger target forward velocity, and a more negative $z_0$ controls a larger target backward velocity. The generated target velocity is insensitive to $z_1 \sim z_3$.

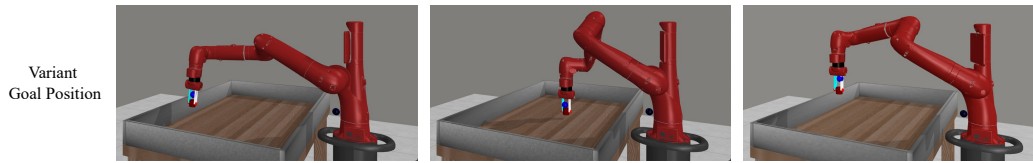

Figure 13: *Pick-Place* experiment's variants

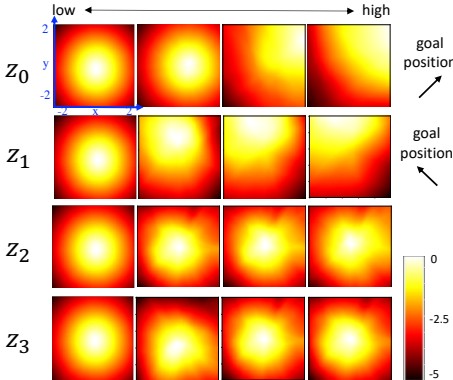

Figure 14: Interpolation on ***Navigation-2D***'s latent context variables. Each row plots the generated tasks' reward heatmaps by assigning from low to high values on the corresponding dimension of latent variable **z** from left to right figures.

## 7.3 ABLATION STUDY

In this section, we present some ablation studies on 1) whether to enforce the disentanglement property of latent context, and 2) whether to introduce physics knowledge to the generative model, 3) the encoder architecture used to do task inference and 4) the components of training loss for encoder-generative model training.

**Ablation study on the encoder structure.** We compare three different architectures:

- A GRU architecture, which is used in our MetaDreamer algorithm.

- A shared multilayer perceptron (MLP) architecture, similar to PEARL Rakelly et al. (2019), which outputs a posterior Gaussian distribution for each transition tuple of the context in parallel and then combines the Gaussians using the standard Gaussian multiplication;

- A multi-head-attention-based architecture, which creates a multi-head key-value embedding for each transition in the context and combines all derived Gaussians for each timestep using standard Gaussian multiplication.

The comparison results on our evaluation metrics are listed in Table 12. MLP-based encoder shows obvious disadvantages in SFI and SCI errors, representing poorer alignment of task inference and generation. The attention-based encoder doesn't obviously underperform our GRU-based encoder, however, requires a good choice of the number of time steps to do Gaussian multiplication. In addition, during meta-testing adaptation, a GRU-based encoder can update the posterior per time-step with the memory of the hidden states in the GRU, while the attention-based encoder needs to keep the memory of all time-step data tuples.

**Ablation study on the components of training loss.** In the paper, we propose to add two cluster losses to help with encoder-generative model learning, intra-cluster loss for encouraging extracting only task-variant-relevant information and inter-cluster loss for task identification and avoiding posterior collapse. We conduct the ablation study to compare training: w/o inter-cluster, w/o intra-cluster, w/o inter- and intra-cluster loss.

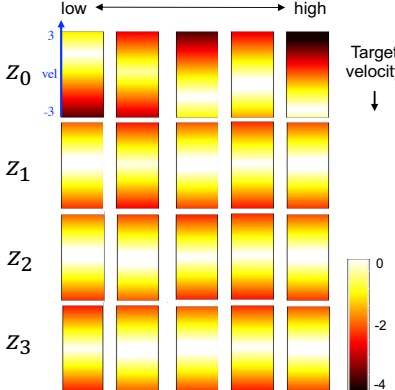

Figure 15: Interpolation on **Walker-2D**'s latent context variables. Each row plots the generated tasks' reward heatmaps by assigning from low to high values on the corresponding dimension of latent variable **z** from left to right figures.

|  | Setting | Disentanglement score | Intra-cluster variance | Reconstruction error | SFI error | SCI error |
|---|---|---|---|---|---|---|
| **MetaDreamer** (ours) | w/ $\beta > 1$ ($\beta$-VAE), w/ physics knowledge, GRU, w/ both cluster losses | $\mathbf{98.3 \pm 0.5\%}$ | $\mathbf{0.36 \pm 0.21}$ | $0.08 \pm 0.01$ | $\mathbf{0.41 \pm 0.23}$ | $\mathbf{0.47 \pm 0.15}$ |
| Disentanglement | w/ $\beta = 1$ (VAE) | $43.1 \pm 5.3\%$ | $1.91 \pm 1.13$ | $\mathbf{0.053 \pm 0.01}$ | $\mathbf{0.27 \pm 0.09}$ | $1.68 \pm 0.43$ |
| Physics | w/o physics knowledge | $92.2 \pm 4.1\%$ | $0.72 \pm 0.18$ | $1.024 \pm 0.02$ | $0.73 \pm 0.12$ | $0.95 \pm 0.26$ |
| Encoder Structure | MLP Attention Model | $94.3 \pm 2.8\%$ $93.7 \pm 3.5\%$ | $0.38 \pm 0.15$ $0.40 \pm 0.17$ | $0.10 \pm 0.03$ $0.09 \pm 0.02$ | $\underline{0.62 \pm 0.27}$ $0.42 \pm 0.18$ | $\underline{0.73 \pm 0.22}$ $0.43 \pm 0.16$ |
| Training Loss | w/o intra-cluster loss w/o inter-cluster loss w/o both cluster losses | $92.7 \pm 2.4\%$ $\underline{72.4 \pm 19.6\%}$ $\underline{68.0 \pm 20.7\%}$ | $\underline{0.73 \pm 0.44}$ $\underline{0.41 \pm 0.16}$ $\underline{0.84 \pm 0.36}$ | $0.08 \pm 0.02$ $\underline{0.24 \pm 0.18}$ $\underline{0.21 \pm 0.13}$ | $0.48 \pm 0.15$ $0.42 \pm 0.17$ $0.51 \pm 0.20$ | $0.71 \pm 0.23$ $0.45 \pm 0.18$ $0.76 \pm 0.27$ |

Table 12: Ablation study results.

[*] This table includes all four ablation studies, each of which is compared with our MetaDreamer, to inspect the settings that we're interested in. Note that the results of the first two ablation studies are from Table 1 in the main paper.
[*] **Bold** means dominant advantages; Underline means dominant disadvantages.

The lower part of Table 12 shows that the intra-cluster loss directly reduces the intra-cluster variance, and the inter-cluster loss benefits disentanglement in learned latent context space by encouraging the discovery of all variants among tasks, and thus further benefits the reconstruction.

## 8    DISENTANGLEMENT SCORE COMPUTATION

We modify the disentanglement metric proposed in Higgins et al. (2016) and illustrate the computation process in Figure 16, and the way to compute the disentanglement score is as follows:

1. Choose a generative factor $\mathbf{g} \sim Unif[1 \dots K]$

2. Randomly choose a reasonable set of generative factors and then enforce different values for the selected generative factor, to consist of a pair of tasks $(\mathcal{T}_1, \mathcal{T}_2)$.

3. Simulate each task and collect corresponding trajectory, $\tau_1^l$ and $\tau_2^l$.

4. Infer $\mathbf{z}_1^l$ with the encoder $q(\mathbf{z}|\tau_1^l)$, and do the same process to get $\mathbf{z}_2^l$.

5. Compute the difference $\mathbf{z}_{\text{diff}}^l = |\mathbf{z}_1^l - \mathbf{z}_2^l|$, the absolute linear difference between the inferred latent representations.

6. Repeat step 2)-5) for $L$ times, and store all inferred latent representation differences.

7. Compute the average difference vector $\mathbf{z}_{\text{diff}}^b = \frac{1}{L} \sum_{l=1}^{L} \mathbf{z}_{\text{diff}}^l$, which is then fed as inputs to a low-capacity linear classifier. The accuracy of this predictor is so-called **disentanglement metric score**

As stated in Higgins et al. (2016), the disentangling metric defined in this way can measure both the independence and interpretability (due to the use of a simple classifier) of the inferred latent variables, while other variety of approaches (such as PCA or ICA) only achieve independence.

Rather than fixing only one generative factor and varying all the others (designed in Higgins et al. (2016)), we, at step 2), propose to fix all generative factors but the selected one. This is because we always set the latent context space with more dimensions than the number of generative factors, thus inactive dimensions may confuse the linear classifier and fail to give the real disentanglement metric score.

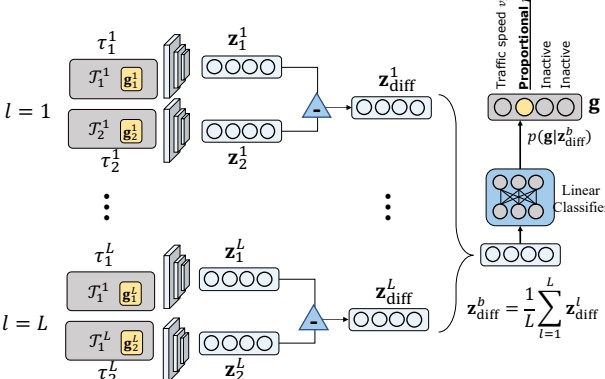

Figure 16: Schematic of the disentanglement metric: over a batch of $L$ samples, each pair of trajectories has only one variant generative factor (here $\mathbf{g}$ = Proportional $p$). We use a pre-train linear classifier to identify the variant generative factor with the average pairwise difference $\mathbf{z}_{\text{diff}}^b$. The probability of identifying the variant generative factor is the derived disentanglement score.

# 9 PSEUDOCODE

The pseudocode of the meta-training phase is described in Algorithm 1. The meta-testing phase is the same as Rakelly et al. (2019), so will not be included here.

# 10 EXPERIMENTAL DETAILS

We give an overview of important hyperparameters and experimental settings we used (Table 13 and 14)

---

**Algorithm 1** MetaDreamer Meta-training

---

**Require:** Encoder $q_\theta$, decoder $p_\phi$, policy $\pi_\psi$, critic $Q_\psi$;

Meta-training tasks $\{\mathcal{T}^{(i)}\}_{i=0,\dots N}$ from task distribution $p(\mathcal{T})$; Replay buffer $\mathcal{B}$ for real data, $\tilde{\mathcal{B}}$ for imaginary data;

**Hyperparameter:** Encoder-decoder learning rates $\alpha_{ED}$, policy learning rates $\alpha_{PR}$ and $\alpha_{PI}$

1: **while** not done **do**
2:     Sampling rollouts for each task $\mathcal{T}^{(i)} \sim p(\mathcal{T})$ and store in $\mathcal{B}^{(i)}$
3:     **for** each encoder-decoder training step **do**
4:         Sample context $\mathbf{c}^{(i)} \sim \mathcal{B}^{(i)}$ (for each task index $i$)
5:         Perform task inference $\mathbf{z} \sim q_\phi(\mathbf{z}|\mathbf{c})$
6:         Calculate loss $\mathcal{L}$ according to Equation 6.
7:         Update encoder-decoder: $(\theta, \phi) \leftarrow (\theta, \phi) + \alpha_{ED}\nabla\mathcal{L}$
8:     **end for**
9:     Do interpolations on the latent context space following Equation 2 to get a set of imaginary latent context $\mathbf{z}^{\mathcal{I}}$, corresponding to imaginary tasks $\mathcal{T}^{\mathcal{I}}$.
10:     Generating imaginary rollouts for each imaginary task $\mathcal{T}^{\mathcal{I}}$ and store in $\mathcal{B}^{\mathcal{I}}$
11:     **for** each policy training step **do**
12:         Sample context $\mathbf{c} \sim \mathcal{B}$ and perform task inference $\mathbf{z} \sim q_\phi(\mathbf{z}|\mathbf{c})$
13:         Sample interpolated latent context $\mathbf{z}^{\mathcal{I}}$ and generate corresponding context $p_\phi(\mathbf{c}^{\mathcal{I}}|\mathbf{z}^{\mathcal{I}})$
14:         Update policy:
        $\psi \leftarrow \psi + \alpha_{PR}(\nabla\mathcal{L}_{critic}(Q_\psi(\mathbf{z}), \mathcal{B}) + \nabla\mathcal{L}_{actor}(\pi_\psi(\mathbf{z}), \mathcal{B})) + \alpha_{PI}(\nabla\mathcal{L}_{critic}(Q_\psi(\mathbf{z}^{\mathcal{I}}), \tilde{\mathcal{B}}) + \nabla\mathcal{L}_{actor}(\pi_\psi(\mathbf{z}^{\mathcal{I}}), \tilde{\mathcal{B}}))$
15:     **end for**
16: **end while**

---

Table 13: Hyperparameters overview.

| Hyperparameter | Value |
|---|---|
| Optimizer | ADAM |
| Discount factor | 0.99 |
| Encoder type | Recurrent network |
| Policy type | Tanh Gaussian policy |
| VAE learning rate | 3e-4 |
| Policy learning rate | 3e-4 |
| Non-linearity (all networks) | Relu |
| Meta batch size | 10 |
| Embedding batch size | Max episodic length |
| Number of updating steps per iteration | 500 |

| Experiment | Highway-merge | Navigation-2D | CartPole | Walker-2D | Pick-Place |
|---|---|---|---|---|---|
| Meta training tasks | 16 | 20 | 30 | 30 | 16 |
| Meta testing tasks | 4 | 5 | 5 | 5 | 4 |
| Maximal episodic length | 300 | 100 | 100 | 300 | 200 |
| Number of considered variations | 1 ($v0$) / 2 ($v1$) | 2 | 2 | 2 | 3 |
| Types of considered variations | Transition Probability | Reward Function | Transition Probability | Transition Probability Reward Function | Reward Function |
| Dimension of latent context space | 4 | 4 | 4 | 4 | 6 |
| VAE encoder network size[1] | (128,64,32) | (64,64,32) | (64,64,32) | (258,128,64,32) | (258,128,128,64) |
| Dynamics decoder size[2] | (128,64,32) | (64,32,16) | (64,32,16) | (256,128,128,64) | (256,128,128,64) |
| Reward decoder architecture | (128,64,32) | (64,32,16) | (64,32,16) | (256,128,64,32) | (256,128,64,32) |
| Posterior sample size (per meta task) | 4500 | 1000 | 1000 | 6000 | 4000 |
| Loss weights: | | | | | |
| - $\alpha_R$ reward reconstruction weight | 1 | 10 | 1 | 5 | 20 |
| - $\alpha_T$ transition reconstruction weight | 100 | 1 | 100 | 50 | 1 |
| - $\alpha_{c1}$ intra-clustering weight | 3 | 1 | 1 | 1 | 1 |
| - $\alpha_{c2}$ inter-clustering weight | 10 | 1 | 2 | 1 | 1 |
| - $\beta$ KL divergence weight | 5 | 2 | 2 | 5 | 5 |

Table 14: Experimental setting overview.

[1] For all experiments, we use a GRU-based encoder, and the size refers to the

[2] This refers to the architecture of the Fully Connected Network (FCN) in the dynamic decoder. Since no physics knowledge is utilized in **Walker-2D** and **Pick-Place**, it is exactly the architecture of the whole dynamics decoder.

