# OpenReview forum: "Dream to Adapt: Meta Reinforcement Learning by Latent Context Imagination and MDP Imagination"
_ICLR.cc/2024/Conference — Submitted to ICLR 2024_

### Official Review · Reviewer_Jvj1 · 2023-10-20

**Soundness:** 2 fair
**Presentation:** 1 poor
**Contribution:** 1 poor
**Rating:** 3
**Confidence:** 4

**Summary:**

This paper proposes to augment meta-RL with task imagination. By learning a better task representation space and interpolating meta-training task encodings, the authors claim to improve over PEARL on several environments.

**Strengths:**

1. Task imagination is an important topic in meta-RL.
2. The idea of integrating Dreamer into meta-RL is interesting.
3. The experiment results show that MetaDreamer is able to greatly outperform PEARL.

**Weaknesses:**

Overall, the paper needs major revision in presentation. The novelty is not clearly presented, and its connection with Dreamer is confusing. Below are my detailed comments.

1. Please use the right style for citations. As instructed in the style files, use \citep to include the citations in parenthesis.
2. Calling the method MetaDreamer is confusing. The major contribution of Dreamer to model-based RL is its ability to directly compute gradients through the dynamics model to optimize policies. However, MetaDreamer seems only to use the model to collect datasets, and does not pass gradients through the model. So it might not be appropriate to call it a meta-RL version of Dreamer.
3. Novelty is not clear. Section 4.1 emphasizes that MetaDreamer uses \beta-VAE. However, \beta-VAE is a common trick used in PEARL, as it also adjusts \beta as a hyper-parameter. Moreover, using GRU as task encoders and decoupling task encoding and policy learning by decoding task rewards and dynamics is also not novel. They are techniques used by VariBAD[1], another SOTA meta-RL algorithm.
4. The presentation needs improvement. The "Meta-imagination by sampling the latent context space" paragraph in Section 4.2 is confusing. I roughly understand that the authors interpolate different dimensions of the task encodings, but Eq.2 is really confusing. I advise the authors to re-construct the math expressions, E.g. the notations $M,K,f(M)$ are not clearly defined.
5. The clustering loss in Section 4.3 is not clearly explained. This clustering loss seems a bit novel, and I think the authors should emphasize this clustering loss in earlier sections like the introduction. There is also a lack of ablation studies to discuss the importance of this loss. Will simply adding this loss to PEARL improve performance?
6. Experiment results are not convincing. Although MetaDreamer seems to outperform PEARL, it utilizes extra physics knowledge of the environment, which is not fair for PEARL and MAML. Also, there lack ablation studies to discuss the importance of MetaDreamer's components. I also expect to see results on more complex tasks like Meta-World.

[1] Zintgraf, Luisa, et al. "Varibad: A very good method for bayes-adaptive deep rl via meta-learning." arXiv preprint arXiv:1910.08348 (2019).

**Questions:**

Please refer to the weakness part. I think a re-organization of the paper, as well as more experimental results will significantly improve the paper.

---

### Official Review · Reviewer_xXsW · 2023-10-25

**Soundness:** 3 good
**Presentation:** 1 poor
**Contribution:** 1 poor
**Rating:** 3
**Confidence:** 5

**Summary:**

The study introduces a Meta-RL algorithm named MetaDreamer, that aims to enhance generalization to unseen tasks. This is achieved through the preliminary training of the agent, utilizing simulated tasks created by interpolating between acquired latent contexts. The performance of MetaDreamer is assessed through its application in scenarios such as a highway-merging task and various MuJoCo tasks. The results are presented with a comparison to baseline performances, indicating a relative improvement in the capabilities of MetaDreamer.

**Strengths:**

**Strength 1.**

The primary innovation of this paper resides in its introduction of physics-guided generative models within the realm of reinforcement learning (RL). This approach necessitates a degree of pre-existing knowledge concerning the dynamics of the tasks at hand, yet it is a reasonable presumption that such information pertaining to physical dynamics is readily accessible for the majority of environments. The incorporation of relevant, physics-informed concepts into machine learning showcases a significant intertwining of these disciplines within the RL framework, highlighting both the novelty and the importance of this development.

**Weaknesses:**

**Weakness 1: Need for Enhanced Clarity and Precision**

This paper would benefit significantly from improvements in clarity across several sections, necessitating comprehensive revision in certain areas.

- The section concerning related work could be more effectively structured. Presently, it poses some challenges in interpretation due to the way the context and citations are presented. A clearer demarcation between different works cited, perhaps through the use of spaces or commas, would greatly enhance readability and comprehension.

- Precision and consistency in the use of notations and equations need attention. Currently, there seems to be some inconsistency in symbols, evidenced by the interchangeable use of $\phi$ and $\theta$ in Equation 1, Equation 4, Figure 2, and the introductory portion of Section 4.3. Clarifying these notations would significantly improve the paper's technical precision.

- Attention to detail could further be improved by rectifying typographical errors scattered throughout the text. An instance of this is the term 'MPD' found on page 8, presumed to be a typo.


**Weakness 2: Clarification of Contribution Required**

The paper would benefit from a more precise delineation of the authors' contributions, particularly within Section 4, which is integral to understanding the novelty of the work presented.

The narrative regarding the unique contributions of this research becomes somewhat obscured, especially in the third paragraph of Section 4. Here, the incorporation of elements such as the GRU encoder and the dynamics decoder, recognized components from VariBAD, alongside the concept of imagined dynamics from LDM, are noteworthy. However, these inclusions call for a clearer distinction between what the present work originates and what it adapts from existing methodologies.

The elements of originality appear to concentrate on the introduction of the disentangled context in Section 4.1 and the physics-informed generative models in Section 4.2. These are indeed significant contributions; however, their impact and the extent of their innovation could be overshadowed by the lack of clear differentiation from prior works.

To strengthen the academic rigor and originality of the paper, it is highly recommended that the authors carefully revise the method section. A more distinct separation between the novel contributions of this paper and the methodologies and concepts borrowed from previous works would enhance the readers' comprehension of the value and innovation this study offers. This clarity is not just a matter of academic precision but also of intellectual honesty and contribution to the field.


**Weakness 3: Areas for Improvement in Empirical Evaluation**

The empirical evaluation section of the paper presents several opportunities for enhancement that would substantially solidify the work's contributions.

Greater emphasis on reproducibility is essential. The current manuscript could significantly benefit from a more detailed presentation of the experimental setup and, importantly, the inclusion of source code. These additions would greatly aid peers in replicating and verifying the results.

The breadth of comparative analysis could be expanded. While the work's assessment includes comparisons with certain methods, the selection seems somewhat narrow. For instance, in the highway merging task and the MuJoCo tasks, the analysis includes MAML and PEARL but would be enriched by considering other recent relevant baselines like VariBAD, which shares considerable similarities with the approach.

The robustness of empirical conclusions in reinforcement learning, and Meta-RL in particular, often hinges on the ability to replicate results across multiple trials. The current reliance on results from three random seeds may not suffice to construct a reliable empirical foundation. It would be highly advisable for the authors to expand this aspect by utilizing a broader range of seeds, perhaps 8 to 10, thereby enhancing the statistical reliability and validity of the conclusions, especially considering the empirical nature of this study.

**Questions:**

Question 1: The autonomous driving experiment showcased in Section 5.1 is indeed inspiring, effectively illustrating an application for the physics-informed decoder. Could you suggest any other benchmarks or scenarios where the proposed method has the potential to be particularly advantageous?

---

### Official Review · Reviewer_iZ95 · 2023-10-31

**Soundness:** 1 poor
**Presentation:** 1 poor
**Contribution:** 1 poor
**Rating:** 1
**Confidence:** 4

**Summary:**

This paper proposes a variant of the PEARL meta-RL algorithm that differs in three ways:
(1) The context variable is computed using a VAE, i.e., a prior is imposed on the context variable, in the hopes that this produces a disentangled representation for the context variable.
(2) Rollouts from the policy can be simulated by knowing the semantic meanings of each dimension of the state and then applying physical laws to get the next state from an application of an action. For example, knowing that certain dimensions of the state represent velocity or acceleration, physical laws are used to relate the two.
(3) Contexts for "imagined" new tasks can be generated by interpolating the context variables computed on existing training tasks.

This work performs experiments on a Highway task and several MuJoCo tasks with the resulting algorithm. The proposed algorithm appears to perform favorably compared to PEARL and MAML.

**Strengths:**

This work attempts to extend PEARL in directions that are quite natural: e.g., to better handle generalization to new tasks, and potentially obtain better context variables.

**Weaknesses:**

This work could make significant improvements in the following areas:

*Presentation*:
- This work is difficult to read and reproduce due to its presentation.
- From a high-level, there are two main issues with the presentation:
- (a) This work does not outline the problems it aims to solve, and hence cannot show that these problems really exist and that the work indeed solves them. Instead, we must rely on results on tasks that have already been previously solved to evaluate the algorithm's contribution (more on this below).
- (b) This work presents its ideas at both extremely high vague levels (e.g., see intro paragraph 1, 2; 4.1: disentangled latent context space; 4.2 first para of "meta-imagination ...") and extremely detailed levels, with nothing in between, leaving the reader to fill in the gaps for what these vague statements precisely mean. The high-level vague claims are often ungrounded or unsubstantiated. The detailed levels also often leave terms undefined or defined with insufficient precision (e.g., f_\theta in Eq (1), Z_k in 4.2 [these seem to be \tau-dim vectors?]; z_{i, j} in Eq (2)). They also appear to be ad-hoc or insufficiently justified. For example, why should interpolating context vectors in Eq (2) be sensible, and in what way do we expect it to be useful?

*Technical rigor / precision / correctness*
- Beyond the notational issues described above, there are also areas that appear to simply be incorrect or misleading.
- In the experiments, why are there modifications to the Walker environment compared to the basic one? The original PEARL paper appears to report stronger results than what this work is reporting on Walker (though it appears to be modified). Additionally, VariBAD reports results that are even better than PEARL. It seems like the VariBAD paper is reporting results that are stronger than what are reported in this paper, though it would be good to see a direct head-to-head comparison.
- Equation (1) appears to be incorrect. I believe that D^{ts} and D^{tr} have been swapped in the equation, where typically you do training on the meta-training dataset D^{tr}, followed by evaluation on the meta-testing dataset D^{ts}, and take your gradients w.r.t. that.
- What is the justification behind making the context variables disentangled with a VAE? Using the context variables to predict the next state and rewards as in VariBAD seems much more principled, since this helps you obtain the sufficient statistics for solving the task. On the other hand, I don't clearly see why we should expect applying a VAE should help with better Thompson-Sampling under a PEARL-like approach.
- Additionally, note that while hard-coding physical laws is a completely reasonable thing to do for generating new simulated trajectories, this seems to be identical to just supporting it under the simulator. This approach also does not nicely generalize to new domains, especially those with pixel inputs, where the dimensions of the state are less easily manipulatable.

**Questions:**

See questions in previous section.

---

### Official Review · Reviewer_eJ5U · 2023-11-07

**Soundness:** 2 fair
**Presentation:** 1 poor
**Contribution:** 3 good
**Rating:** 3
**Confidence:** 4

**Summary:**

The submission proposes a meta-RL framework, MetaDreamer, designed to improve generalization and sample efficiency compared to previous works.
MetaDreamer learns disentangled task latent space and generative models of trajectories, then uses them to augment new tasks and trajectory data for each task.
Physics-related inductive bias is imployed to improve the training of the task encoder and decoder models.
Experimental results suggest that MetaDreamer demonstrates better sample efficiency and generalization performance than previous meta-RL methods in Highway-merging and Mujoco control domains.

**Strengths:**

- The idea of augmenting both tasks and trajectory data has not been explored in previous works and is technically challenging.
- Experimental results demonstrate promising sample efficiency and generalization capabilities compared to existing meta-RL methods.

**Weaknesses:**

**Essential baselines are missing**
The provided set of experiments only presents evaluation results for PEARL and MAML without task interpolation or physics-informed encoder-decoder structure.
While the potential benefits of the proposed model over existing works involving task interpolation (ex. LDM [1]) or task encoding using inductive bias (ex. VariBad [2] + physics-informed encoder-decoder structure) are intuitively understandable, there is a lack of empirical comparisons.
Furthermore, as the proposed method incorporates several incremental modification from previous works (beta-VAE, task interpolation, data augmentation, and physics-based inductive bias), it remains unclear which specific modification or combination thereof contributes to the observed improvement over meta-RL methods without those modifications.

**Presentation has room for improvement**
Some flaws of writings make hard to understand their approach:
- Figures 2 lacks explanations for indicators 1 to 3.
- Section 4.2 notations lack critical details, thus hard to follow. For examples :
  - $Z_i$ is not defined.
  - Notation in $z_{f(k), i-1}$ is not explained.
- The approach to the physics-informed generative model is not sufficiently explained in any part of the paper. Section 4.2 provides limited detail and focuses on one specific domain.
- Section 1 - "... without additional test-time adaptation..." - seems to be inaccurate on my understanding. The proposed work has a same test-time adaptation procedure as PEARL, and the imagination is done during meta-training.
- In Figure 7,
  - the final performance does not align with actual final value of each graph
  - it is not clear how many runs were averaged for each graph
- Citation formatting substantially harms readability

[1] Improving Generalization in Meta-RL with Imaginary Tasks from Latent Dynamics Mixture. Lee and Chung. NeurIPS 2021.
[2] VariBAD: A Very Good Method for Bayes-Adaptive Deep RL via Meta-Learning. Zintgraf at al. ICLR 2020.

**Questions:**

It appears that the Walker-2D domain may be susceptible to compounding errors when generating 300-timestep length rollouts from initial states. Would it be possible to provide a qualitative visualization of the generated observations from the Walker-2D domain to show whether the generated observations align with provided tasks?

---

### Meta-Review · Area_Chair_LCpS · 2023-12-06

**Metareview:**

The paper presents a novel model-based algorithm for meta learning, as well as a domain-specific decoder architecture, and evaluates the method in a state-based driving domain and some continuous control tasks. While ideas in the paper seem interesting, reviewers agree that the method is not presented clearly and the empirical evaluation is lacking. The authors did not respond to the reviews.

**Justification For Why Not Higher Score:**

Lack of clarity, unconvincing experimental results, misleading/unscientific plot axes

**Justification For Why Not Lower Score:**

N/A

---

### Decision · Program_Chairs · 2024-01-16

Reject